# Targeting enhancer switching overcomes non-genetic drug resistance in acute myeloid leukaemia

Charles C. Bell [1,2,12], Katie A. Fennell[1,2,12], Yih-Chih Chan [1,2], Florian Rambow[3], Miriam M. Yeung [1],
Dane Vassiliadis [1], Luis Lara[1,2], Paul Yeh[1,2,4], Luciano G. Martelotto[5], Aljosja Rogiers[3,6], Brandon E. Kremer[7],
Olena Barbash[7], Helai P. Mohammad[7], Timothy M. Johanson[8,9], Marian L. Burr [1,2], Arindam Dhar[7],
Natalie Karpinich[7], Luyi Tian [8,9], Dean S. Tyler[1,2], Laura MacPherson[1,2], Junwei Shi [10], Nathan Pinnawala[1,2],
Chun Yew Fong[1,2], Anthony T. Papenfuss[1,8,9], Sean M. Grimmond [5], Sarah-Jane Dawson [1,2,5],
Rhys S. Allan [8,9], Ryan G. Kruger[7], Christopher R. Vakoc[11], David L. Goode[1,2], Shalin H. Naik [8,9],
Omer Gilan[1,2], Enid Y.N. Lam [1,2], Jean-Christophe Marine[3,6], Rab K. Prinjha [7] & Mark A. Dawson [1,2,4,5]

Non-genetic drug resistance is increasingly recognised in various cancers. Molecular insights into this process are lacking and it is unknown whether stable non-genetic resistance can be overcome. Using single cell RNA-sequencing of paired drug naïve and resistant AML patient samples and cellular barcoding in a unique mouse model of non-genetic resistance, here we demonstrate that transcriptional plasticity drives stable epigenetic resistance. With a CRISPR-Cas9 screen we identify regulators of enhancer function as important modulators of the resistant cell state. We show that inhibition of Lsd1 (Kdm1a) is able to overcome stable epigenetic resistance by facilitating the binding of the pioneer factor, Pu.1 and cofactor, Irf8, to nucleate new enhancers that regulate the expression of key survival genes. This enhancer switching results in the re-distribution of transcriptional co-activators, including Brd4, and provides the opportunity to disable their activity and overcome epigenetic resistance. Together these findings highlight key principles to help counteract non-genetic drug resistance.

[1] Cancer Research Division, Peter MacCallum Cancer Centre, Melbourne, VIC, Australia. [2] Sir Peter MacCallum Department of Oncology, University of Melbourne, Melbourne, VIC, Australia. [3] Laboratory for Molecular Cancer Biology, VIB Center for Cancer Biology, KU Leuven, Leuven, Belgium. [4] Department of Haematology, Peter MacCallum Cancer Centre, Melbourne, VIC, Australia. [5] Centre for Cancer Research, University of Melbourne, Melbourne, VIC, Australia. [6] Department of Oncology, KU Leuven, Leuven, Belgium. [7] Epigenetics DPU, GlaxoSmithKline, Collegeville, Pennsylvania, USA. [8] The Walter and Eliza Hall Institute of Medical Research, Melbourne, VIC, Australia. [9] The Department of Medical Biology, The University of Melbourne, Melbourne, VIC, Australia. [10] Perelman School of Medicine, University of Pennsylvania, Philadelphia, PA 19104, USA. [11] Cold Spring Harbor Laboratory, Cold Spring Harbor, New York, USA. [12] These authors contributed equally: Charles C. Bell, Katie A. Fennell. Correspondence and requests for materials should be addressed to M.A.D. (email: Mark.Dawson@petermac.org)

Genetic clonal evolution in cancer cells has undisputedly been demonstrated to play a central role in mediating resistance to a range of targeted and conventional chemotherapies[1–5]. However, therapeutic resistance in the absence of a clear genetic cause is increasingly being recognized in several cancers including acute myeloid leukaemia (AML)[6–11]. Clinical experience over many decades shows that when patients relapse after achieving a remission to an anti-cancer therapy, re-challenging those patients with the same therapy is invariably futile. This is despite the fact that these patients have not been exposed to the anti-cancer therapy for several months. Not infrequently, genomic analyses fail to identify a genetic cause for this stable drug resistance[1–11] and even when new coding mutations are identified, rarely have they been proven to mediate the resistance phenotype.

The spontaneous mutation rate in AML is low and despite the analysis of thousands of AML genomes, there is no clear evidence of pervasive genomic instability or a hypermutator phenotype. Several independent groups have surveyed the genomes of paired pre-therapy and relapse AML cells[1–4] and these data, collected from both adult and paediatric patients, show that acquired resistance emerges without new non-synonymous coding mutations in up to 40% of patients within some cohorts (Fig. 1a). Interestingly, resistance in the absence of new mutations is not significantly associated with reduced time to relapse suggesting that this is unlikely to represent an unrecognized form of genetically driven primary resistance (Supplementary Fig. 1A).

We have previously shown in a murine AML model that acquired therapeutic resistance to BET inhibitors, an epigenetic therapy capable of inducing complete remissions in AML patients[12,13], emerges in the absence of new genetic mutations from leukaemia stem cells (LSC)[6]. Importantly, we demonstrated that this form of non-genetic resistance was stable, as cells that were re-challenged after prolonged drug withdrawal remained resistant to the therapy[6]. Using this unique AML model, which recapitulates many of the features of non-genetic resistance in human AML, we sought to establish some of the key principles involved in this process.

Here we observe that clinical resistance to BET inhibitors can arise in the absence of new genetic mutations. Using molecular barcoding and single cell transcriptomics in a unique mouse model, we show that stable non-genetic resistance to BET inhibitors is acquired through dynamic transcriptional adaptation. We identify regulators of enhancer formation as key mediators of the resistant state and show that non-genetic therapy resistance can be overcome by switching the enhancer dependencies of key genes required for cell survival. As new enhancer formation underpins non-genetic resistance to cancer therapies, these findings provide a molecular rationale to use epigenetic drugs as maintenance therapies to negate non-genetic transcriptional adaptation.

## Results

### Clinical evidence of non-genetic resistance to BET inhibitors.
To explore the clinical relevance of our previously established findings in a mouse model of BET inhibitor resistance, we analysed serial bone marrow samples taken from two patients (Fig. 1b, c) enrolled on the Molibresib, Phase-1 trial in AML[13]. Genomic analyses failed to reveal the emergence of any new AML mutations or a selective advantage to pre-therapy clones at clinical relapse (Supplementary Fig. 1B). Consistent with these data, mathematical modelling of the kinetics of disease progression reinforce that it is highly unlikely that therapeutic resistance is due to acquired genomic evolution (Supplementary Fig. 1C, D and Supplementary Note 1 and Supplementary Fig. 11). In the

absence of a clear genetic cause for resistance, we sought to explore the adaptive transcriptional responses seen in these patients using single cell mRNA sequencing (scRNA-seq).

These data allowed us to make several key observations: as shown in patient BET001, the transcriptional program associated with resistance was observed in the residual cells at the time of best clinical response. Once established, this adaptive program was conserved and enabled the cells to expand in the context of ongoing drug treatment resulting in clinical relapse (Fig. 1b). Interestingly, as shown in patient BET002, even once therapy had been withdrawn, the malignant cells that survived the therapeutic challenge did not revert to the transcriptional state of the pre-therapy population (Fig. 1c). Instead, they maintained a distinct phenotypic and transcriptional profile, which shared several similarities with human LSC[14,15] (Fig.1d, e, Supplementary Fig. 2A). Consistent with the likelihood of non-genetic adaptation, these marked transcriptional changes were observed in cells harboring the same driver mutation in the pre-therapy and resistant state (Fig. 1f). Although these data are derived from a limited number of patients, they recapitulate many of the key findings derived from our AML mouse model[6]. Moreover, they are also consistent with recent efforts, which have demonstrated that resistance to conventional chemotherapy also emerges from malignant cells with LSC properties without any new gene mutations[8].

### Non-genetic resistance emerges *via* transcriptional adaptation.
While epigenetic resistance is seen clinically to both targeted and conventional therapies, it remains unclear if the transcription programs that facilitate therapeutic evasion are already present in a pre-existing sub-population of AML cells or if they are acquired as an adaptive response to drug pressure. Addressing this question requires a malleable ex vivo model that recapitulates the emergence of therapeutic resistance from LSC in the absence of genetic evolution. Our unique model of BET inhibitor resistance reiterates many of the clinical features described above and provides an ideal system to study the major molecular principles involved in this process.

To understand if resistance was derived from a rare pre-existing population of cells harboring the resistant LSC transcription program prior to drug exposure, we DNA barcoded drug naïve cells and regenerated the resistant population as previously described (Fig. 2a)[6]. These data show that only a small proportion of transformed cells, grown in liquid culture, have sustained clonogenic potential in methylcellulose (Fig. 2b). Notably, however, when increasing therapeutic pressure is applied to cells with clonogenic potential, we found that at least 10% of these malignant clones are able to mount an adaptive response that enables their survival (Fig. 2b, Supplementary Fig. 2B). We also coupled our cellular barcoding studies with concurrent scRNA-seq to gain further insight into the adaptive process. Here we performed scRNA-seq using Cel-Seq2 on the drug naïve cells with clonogenic potential as they were serially re-plated in either vehicle or increasing concentrations of drug, until we derived cells that were resistant to 1000 nM of IBET (>IC90)[6]. Analysis of the scRNA-seq data from cells sampled at each stage of the drug resistance process allowed us to monitor the transcriptional trajectories of individual cells. Pseudo-temporal ordering and t-SNE analysis of these data shows that therapeutic evasion is due to dynamic transcriptional adaptation rather than selection for a rare pre-existing transcriptional program (Fig. 2c, Supplementary Fig. 2C). Together, these data provide compelling evidence for dynamic transcriptional plasticity as a major component contributing to non-genetic therapeutic resistance.

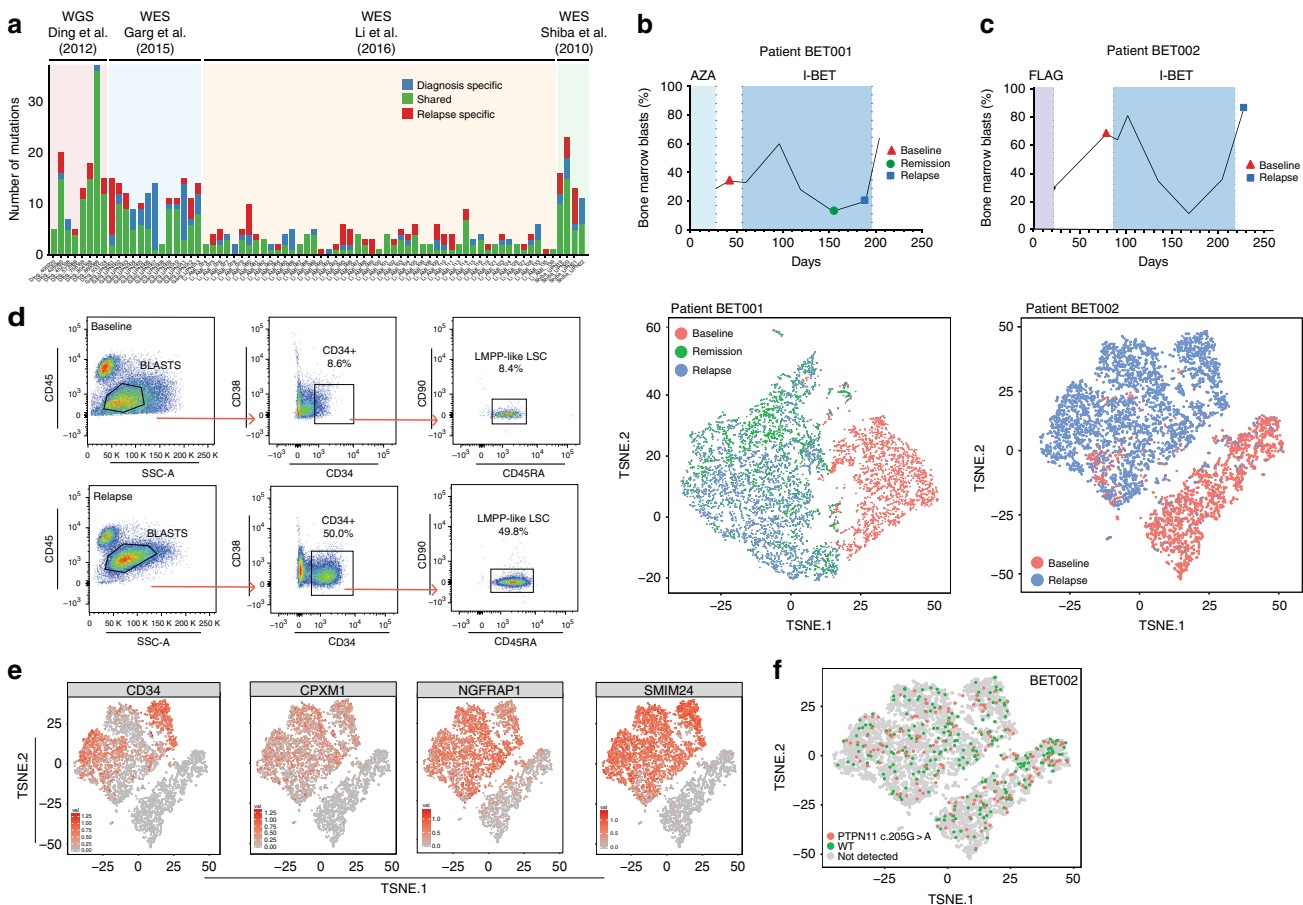

**Fig. 1** Non-genetic adaptation drives clinical resistance in AML. **a** Meta-analysis from four independent studies analysing either the whole genome or whole exome of AML patients at diagnosis and relapse. Mutations are defined as non-synonymous changes within the coding sequence of any gene. Shared mutations are mutations present at both diagnosis and relapse. Whole exome sequencing data from Li et al. (REF 4) was analysed to access the mutations in known AML genes, as defined by the authors. **b** Schematic of the treatment regime and bone marrow blast percentage for patient BET001 over the clinical trial treatment course (top panel). t-SNE analysis of 7360 individual blast cells isolated from patient BET001 at baseline, remission and relapse (bottom panel). scRNA-seq and genomic DNA sampling points are highlighted on the schematic. **c** Schematic of treatment regime and bone marrow blast percentage for patient BET002 over the clinical trial treatment course (top panel). t-SNE analysis of 6349 single blast cells isolated from patient BET002 at baseline and relapse (bottom panel). scRNA-seq and genomic DNA sampling points are highlighted on the schematic. **d** Flow cytometry analysis of cells from patient BET002 at baseline and relapse identifies enrichment for LMPP-like LSCs at relapse based on CD34 + CD38-CD90-CD45RA + expression. Gating strategy is defined by boxes. **e** Expression analysis of selected LSC signature genes (defined in REF 15) in blast cells from patient BET002 overlaid onto the t-SNE plot. **f** Expression of mutant *PTPN11* (c.205 G > A) and WT *PTPN11* in blast cells from patient BET002 overlaid onto the t-SNE plot. Mutant *PTPN11* transcripts were detected in 70 baseline blast cells and 112 relapse blast cells. WGS whole genome sequencing. WES whole exome sequencing, WGS whole genome sequencing, WES whole exome sequencing, AZA azacitidine, FLAG fludarabine, cytarabine and G-CSF

Epigenetic phenotypes in development and disease have been well established to be stably inherited by daughter cells[16,17], and we have shown that these cells were unperturbed by therapeutic re-challenge after prolonged withdrawal from drug (Supplementary Fig. 3A). To investigate this process further, both bulk RNA-seq and scRNA-seq was performed on resistant cells withdrawn from the drug. Principal component analysis of these data show that the transcriptional profile of the resistant cells maintained on drug and resistant cells withdrawn from drug cluster closer together and away from the transcriptional program of the drug naïve cells (Supplementary Fig. 3B–C). Interestingly, although IBET influences the expression of many genes in the cells maintained on drug, the stable resistance program remains unperturbed upon drug exposure (Fig. 2d). Importantly, our scRNA-seq data in these cells shows that there is no spontaneous reversion of even a small subset of resistant cells back to the transcriptional program of drug naïve cells, suggesting that there is no flux between the populations, further supporting the fact that the resistant transcriptional state is stable (Fig. 2e).

To assess the generalisability of our findings to other cancers and therapies, we treated several patient-derived xenograft (PDX) models of cutaneous melanoma with a BRAFV600E/MEK-inhibitor combination until resistance developed. In these models, we clearly observe representative examples of intrinsic and acquired resistance (Supplementary Fig. 3D–F). In the MEL029 model, a pre-existing *PI3KCA* mutation drives intrinsic resistance to combination therapy (Supplementary Fig. 3D). However, all treated lesions from the MEL015 and MEL006 models achieve a dramatic and durable response to this therapy[18]. Similar to the well-described clinical course, we find that continuous MAPK-inhibition ultimately results in acquired resistance in all cases (Supplementary Fig. 3D–F). In MEL015, we find several clinically prevalent acquired mutations[11,19] to account for the resistance phenotype (Supplementary Fig. 3E). Remarkably, however, MEL006 is a representative example where stable resistance emerges in the absence of a resistance-conferring mutation (Supplementary Fig. 3F). These data are consistent with the reported clinical findings in melanoma patients treated with

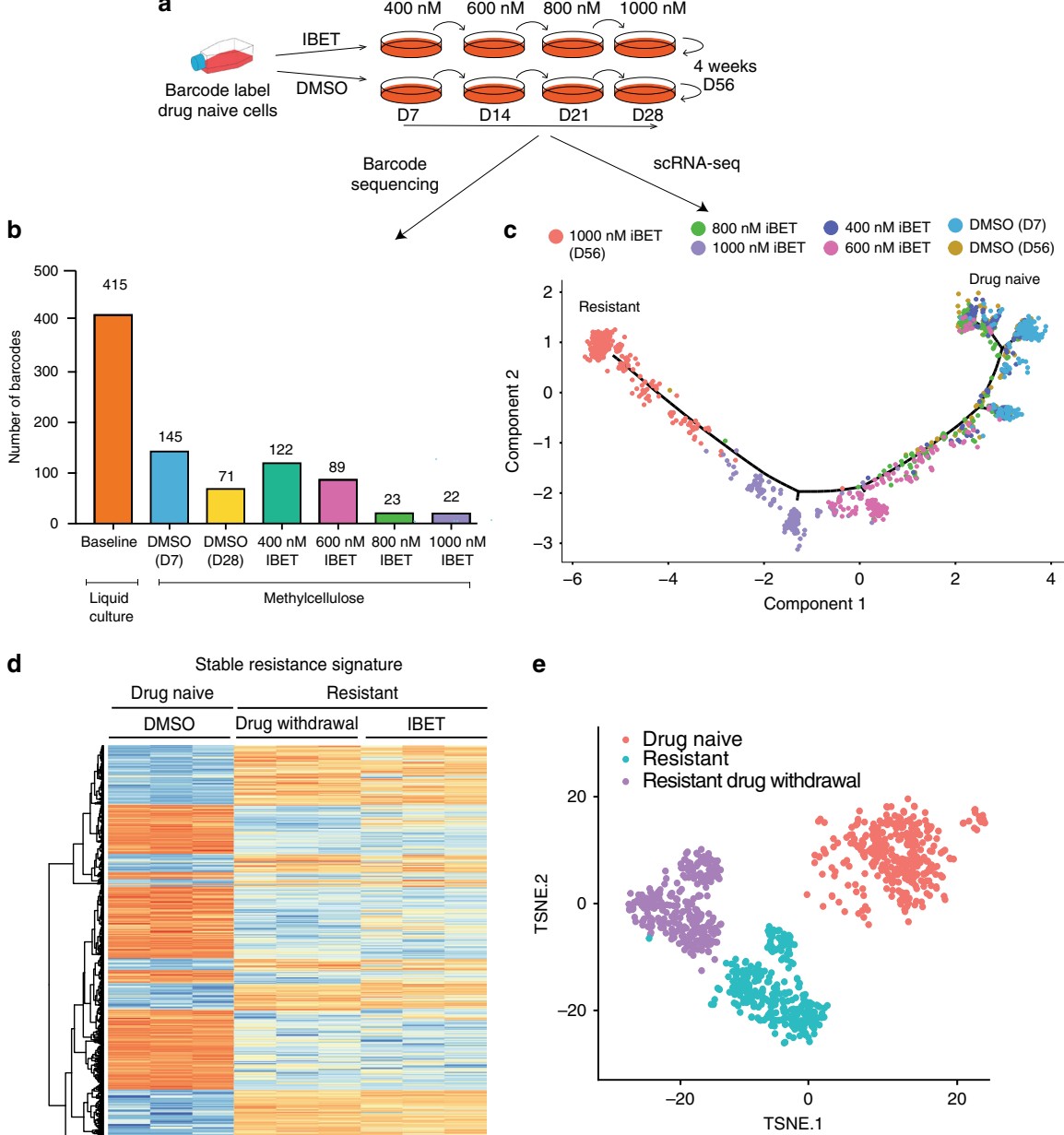

**Fig. 2** Transcriptional plasticity underpins stable non-genetic resistance to BET inhibitors. **a** Schematic highlighting experimental design for following the acquisition of IBET resistance. Drug naïve cells were barcoded in liquid culture and plated in methylcellulose. Cells were then re-plated weekly in either DMSO or escalating concentrations of IBET until they were growing in 1000 nM, where they were maintained for an additional 4 weeks. **b** Number of barcodes comprising 90% of total reads at first plating that are then serially retained at 90% of the total reads for every re-plating either in DMSO or IBET. Representative of two biological duplicates. **c** Pseudotemporal ordering of single drug naïve cells serially passaged in DMSO or increasing concentrations of IBET, as defined by the Monocle2 algorithm. Analysis was conducted based on differentially expressed genes between resistant versus drug naïve cells from bulk RNA-sequencing. **d** Bulk RNA-seq heatmap displaying the expression of genes from the stable resistance signature in drug naïve, resistant, and resistant cells withdrawn from IBET. This subset of genes were determined from finding the differentially expressed genes between drug naïve cells and resistant cells withdrawn from IBET for 6 days. **e** t-SNE analysis of 979 drug naïve cells, resistant cells and resistant cells withdrawn from IBET for 4 days

MAPK inhibitors[11] and highlight the fact that non-genetic resistance is not confined AML or epigenetic therapies.

**Stable non-genetic resistance can be overcome**. Although many examples of non-genetic resistance to cancer therapies have been recently described, most of these studies deal with what has been termed a 'drug persister' state. Persistence is a state of reduced growth and altered metabolism that enables the cell to have a higher tolerance to therapeutic pressure[20–25]. Importantly, in many cases, these persister cells spontaneously revert to the drug sensitive state following drug withdrawal. Therefore, many drug persisters can be eliminated following re-challenge after a drug holiday[20–25]. These cases differ from our model and clinical experience in AML, where resistance is not overcome by a drug holiday. It is largely unclear if and how stable non-genetic resistance can be overcome.

In our AML model, we have previously demonstrated that the majority of resistant cells adopt an immature phenotype lacking markers of myeloid differentiation including Gr1, Cd11b and

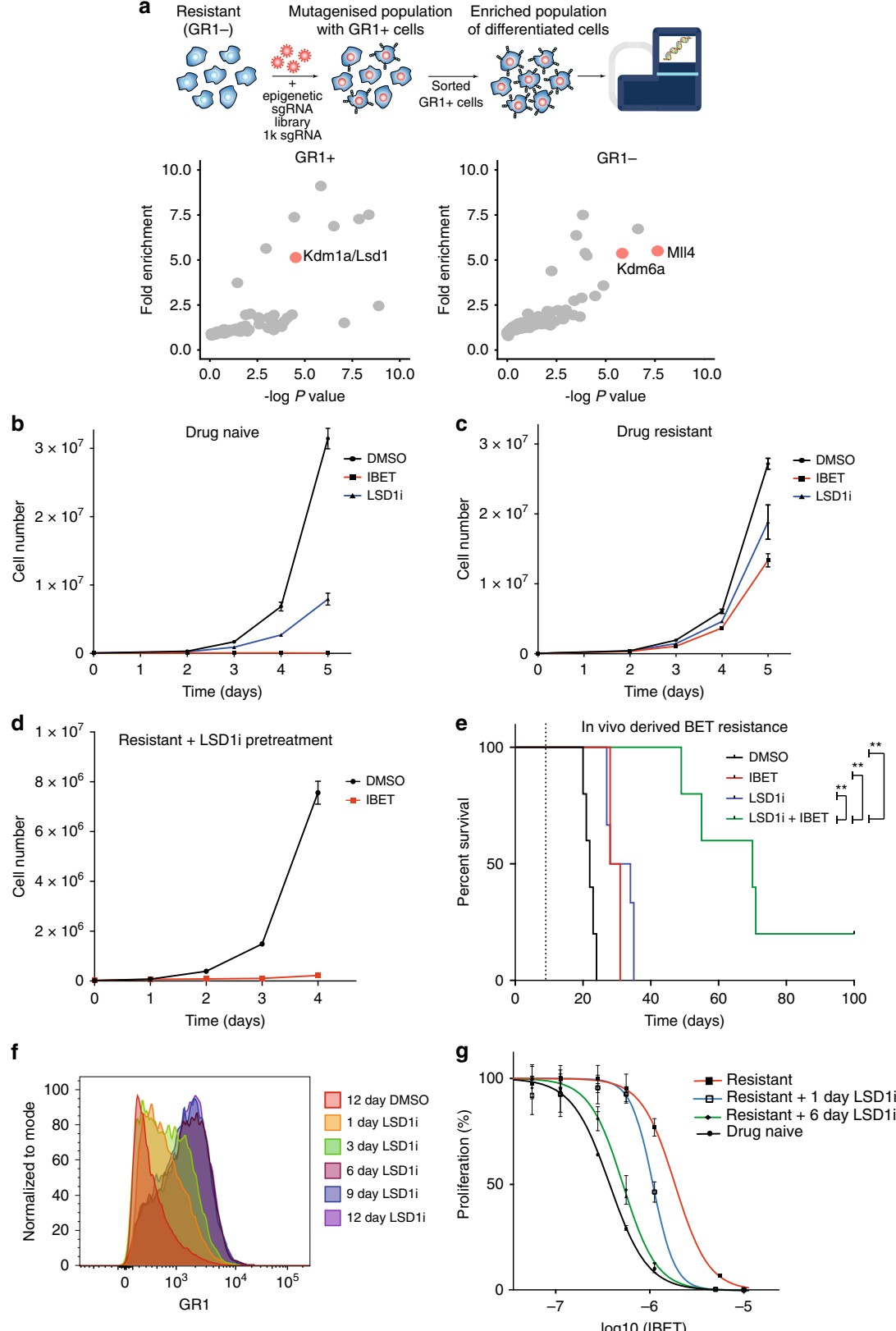

Cd86[6]. Therefore, to understand if epigenetic reprogramming could reverse the stable epigenetic resistance, we performed a focused CRISPR/Cas9 screen[26] to identify the chromatin modulators required to maintain this immature phenotype (Fig. 3a, Supplementary Data 1). These data show that sgRNAs

for *Mll4* (*Kmt2d*) and the associated complex member *Utx* (*Kdm6a*) are enriched in Gr1− cells and consequently prevent differentiation of the resistant population. In contrast, sgRNAs against *Lsd1* (*Kdm1a*) were enriched in cells that adopt a mature Gr1 + state (Fig. 3a and Supplementary Data 1). As both the

**Fig. 3** Inhibition of Lsd1 overcomes stable drug resistance in vitro and in vivo. **a** Epigenetic protein domain-focused CRISPR-Cas9 screen designed to identify epigenetic proteins that sustain the immature Gr1- resistant immunophenotype. Gr1 + cells were enriched by FACS 7 days after sgRNA transduction. GR1— enrichment was determined by the depletion of guides in the GR1 + population. Epigenetic proteins that regulate the enhancer modification H3K4me1/2 are highlighted in red. **b** Proliferation assay of drug naïve and (**c**) drug resistant cells treated with DMSO, IBET (1000 nM) or GSK-LSD1 (500 nM). Error bars represent S.E.M of 3 cell culture replicates. Representative of 3 biological replicates. **d** Proliferation assay of drug resistant cells pre-treated for 6 days with GSK-LSD1 (500 nM) followed by treatment with DMSO or IBET (1000 nM). Error bars represent S.E.M of 3 cell culture replicates. Representative of 3 biological replicates. **e** Kaplan–Meier curve of vehicle and drug treated mice transplanted with MLL-AF9 leukaemic cells serially re-transplanted (4 generations) in the presence of IBET treatment (20 mg/kg) to enrich for IBET-resistant leukaemia. $n = 6$ mice per group. Dotted line indicates the start of treatment. Dosing was performed by IP injection once a day at 20 mg/kg for IBET-151 and/or 0.5 mg/kg for GSK-LSD1. Log rank (Mantel-Cox) for DMSO versus LSD1i + IBET: $p < 0.002$, IBET versus LSD1i + IBET: $p < 0.002$ and LSD1i versus LSD1i + IBET: $p < 0.002$. **f** Flow cytometry analysis of Gr1 surface expression after treatment of drug resistant cells with GSK-LSD1i (500 nM) for the indicated durations. **g** Dose–response assay (IC50) to IBET of drug naïve, resistant cells and resistant cells pre-treated with GSK-LSD1i (500 nM) for either 1 or 6 days. Error bars represent S.E.M of 4 cell culture replicates

---

methyltransferase Mll4-Utx complex and demethylase Lsd1 functionally antagonize each other to regulate enhancer activity via methylation of histone H3K4 (H3K4me1/2)[27], these data raised the possibility that enhancers may be key mediators of the resistance phenotype. We validated the importance of Lsd1 in driving the mature phenotype using RNA interference (RNAi) (Supplementary Fig. 4A, B) and a selective Lsd1 catalytic inhibitor[28] (Supplementary Fig. 4C).

To understand the therapeutic implications of these findings, we treated the drug naïve and resistant cells with an Lsd1 inhibitor. Consistent with previous reports, we find that inhibition of Lsd1 is sufficient to attenuate the growth of the drug naïve AML population[29,30], however it had little effect on the growth of the resistant cells (Fig. 3b, c). Remarkably, we noted that although Lsd1 inhibition is ineffective at reducing the growth of the resistant cells, it restored their sensitivity to BET inhibition (Fig. 3d). To further assess the relevance of these findings in vivo, we serially passaged drug naïve leukaemia cells in mice with therapy to increase their refractoriness to BET inhibition[6]. As a control, we similarly passaged a matched population of AML cells without drug exposure in vivo. In accordance with published data, Lsd1 inhibition (LSD1i) was effective in extending survival in mice transplanted with drug naïve AML cells[29,30] (Supplementary Fig. 4D). However, in line with our data ex vivo, we found that LSD1i was largely ineffective against the in vivo drug adapted cells but it was able to re-instate the in vivo sensitivity to BET inhibition (Fig. 3e). Importantly, the resensitisation to BET inhibition in vitro is not due to an immediate response to combination therapy, instead it is a time-dependent process that requires several days to be effective (Fig. 3f, g and Supplementary Fig. 4E). Consistent with these findings, there is little evidence of synergy with the combination of Lsd1 and BET inhibitors in a broad range of human AML cells (Supplementary Fig. 4F). Together these findings suggest that LSD1i reinstates sensitivity to BET inhibition through a time-dependent modulation of the resistant cells.

**LSD1i restores sensitivity to IBET via new enhancer formation.** Consistent with the immunophenotypic changes induced by LSD1i treatment, the global transcriptome analyses of the resistant cells following LSD1i treatment showed an increased expression of a myeloid differentiation program and down-regulation of the LSC signature (Supplementary Fig. 5A, B). Notably however, although the resensitised cells are immunophenotypically similar to the drug naïve population, the overall transcriptional program of these cells was quite distinct, suggesting that LSD1i treatment did not simply result in the restoration of the transcriptional program of the drug naïve cells (Supplementary Fig. 5C). Therefore, we investigated whether the specific gene expression changes associated with resistance to IBET were reversed by LSD1i treatment. These analyses showed

that very few of the gene expression changes associated with IBET resistance, including the previously described the Wnt/β-catenin signature[6], were reversed by LSD1i (Supplementary Fig. 5D, E). Similarly, although treatment of the resistant cells with LSD1i led to substantial gene-expression changes, few of these genes were responsive to IBET and gene ontology analyses did not reveal any significant pathway that accounted for the renewed sensitivity to IBET following LSD1i treatment (Supplementary Fig. 5F). Together, these data demonstrate that resensitisation following LSD1i treatment could not be explained by a reversion of the transcriptional program associated with resistance.

We and others have shown that treatment of drug naïve cells with BET inhibitors results in the potent and rapid repression of a set of broadly expressed genes such as *Myc*, *Bcl2* and *Cdk6* that are essential for the survival of cancer cells (Fig. 4a)[31–34]. In the resistant cells, the expression of these genes is unaffected by IBET (Fig. 4a) and interestingly, our scRNA-seq data show that these genes are sustained in expression throughout the adaptive response to IBET (Supplementary Fig. 6A). Consistent with the lack of efficacy of Lsd1 inhibition in the resistant cells, the expression of this set of genes is unaltered by Lsd1 inhibitor treatment (Fig. 4a). Surprisingly however, we found that although Lsd1 inhibition does not affect their expression, it re-instates the ability of BET inhibitors to repress this set of broadly expressed essential genes, explaining why Lsd1 inhibition is able to overcome the resistance phenotype (Fig. 4a and Supplementary Fig. 6B–C).

As our CRISPR screen suggested that regulation of enhancers appeared to be a central process required for resensitisation, we performed ATAC-seq to assess differential chromatin accessibility sites in the drug naïve cells, and resistant cells before and after LSD1i treatment. Although, LSD1i treatment resulted in widespread changes in chromatin accessibility, we focused our attention around the set of genes repressed by BET inhibition in the drug naïve cells, as these genes are key to cancer cell survival and their responsiveness to BET inhibitor is reinstated in resistant cells pretreated with LSD1i (Fig. 4a). Around these genes, Lsd1 inhibition resulted in the formation of entirely new putative cis-regulatory elements that are not present in the drug naïve or resistant population (Fig. 4b). These new sites of increased chromatin accessibility contain the hallmark chromatin modifications of active enhancers including H3K4me1/2 and H3K27ac (Fig. 4b, Supplementary Fig. 7A). Interestingly, we noted that following Lsd1 inhibition, Brd4 and its associated co-factor Med1 were redistributed from the enhancers present in the resistant cells to these newly formed enhancers (Fig. 4c, d). Using Click-seq[35], which identifies regions of the genome where Brd4 is bound via its bromodomains and hence predicts responsiveness to BET inhibition; we demonstrated that the increased Brd4 occupancy at these new enhancers is bromodomain dependent (Supplementary Fig. 7B). Consequently, Brd4 is displaced from these loci with IBET treatment, resulting in the renewed

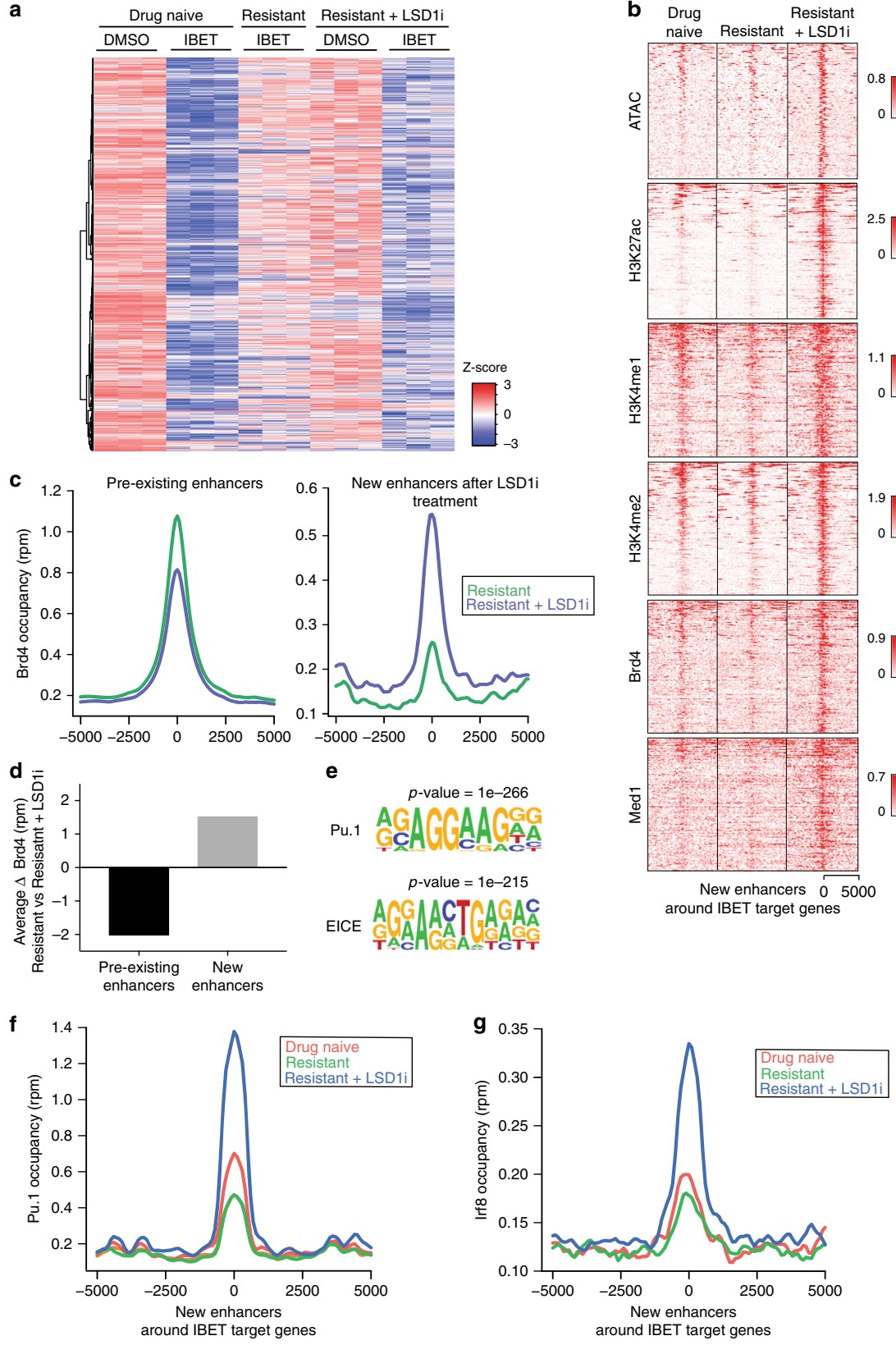

repression of this critical broadly expressed set of genes (Fig. 4a and Supplementary Fig. 7C).

**Differentiation is not sufficient for resensitization.** Although Lsd1 inhibition results in the formation of new enhancers around

these broadly expressed genes required to maintain the proliferation and survival of the cancer cells, the transcriptional output of these genes remains stable (Fig. 4a, b). These intriguing findings are consistent with a form of enhancer switching[36,37], which has been proposed to maintain the expression of important broadly

**Fig. 4** New enhancer formation overcomes therapeutic resistance by reestablishing a targetable dependency. **a** RNA-seq heatmap displaying the response of genes downregulated in drug naïve cells after 6 h of IBET (1000 nM) treatment. This core subset of genes underpins the functional effects of IBET that result in the loss of viability of the drug naïve cells. Also shown here is the expression of these genes in cells stably resistant to 1000 nM of IBET and resistant cells treated for 6 days with GSK-LSD1i (500 nM) followed by treatment with DMSO or IBET (1000 nM) for 6 h. **b** Heatmap of chromatin accessibility and ChIP-seq occupancy for the specified proteins at the newly activated enhancer elements that form after 6 days GSK-LSD1i (500 nM) treatment of the resistant cells. These new enhancer elements are within ($+/-$) 50 kb of the transcriptional start site (TSS) of the genes shown in (**a**). New enhancers are defined by ATAC-seq peaks that show a > 4-fold increase in H3K27ac. The heatmap is centered to display 5 kb either side of the ATAC-seq summit. **c** Average profile of Brd4 occupancy at pre-existing and new enhancers for the genes shown in (**a**). Pre-existing enhancers are defined as H3K27ac peaks in the resistant cells within 50 kb of the TSS of the genes in (**a**). New enhancers are sites within 50 kb of the TSS of the gene that show a > 4-fold change increase in H3K27ac after cells are treated for 6 days with GSK-LSD1i (500 nM). **d** Quantification of change in Brd4 occupancy at pre-existing and new enhancers displayed in (**c**). **e** De novo motif analysis of all newly activated enhancers (ATAC-seq peaks with > 4-fold increase in H3K27ac) after 6 days GSK-LSD1i (500 nM) treatment. **f** Average profile of Pu.1 and **g** Irf8 chromatin occupancy at the newly activated enhancer elements shown in (**b**) in drug naïve, resistant and resistant cells treated for 6 days with GSK-LSD1i (500 nM)

expressed genes throughout development, to cater for changes in the specific transcription factor repertoire (Supplementary Fig. 8A). A key feature of enhancer elements is their ability to provide sequence specific binding platforms for the coordinated function of transcription factors[27,38]. Therefore, to explore the underlying basis for the specific regulation of these new enhancers, we asked which transcription factor binding sites were most significantly enriched at the new enhancers formed after LSD1i treatment. The most significant transcription factor motifs were a Pu.1 motif and a composite EICE motif involving the lineage defining pioneer factor Pu.1 and the transcription co-factor Irf8 (Fig. 4e)[39]. Although rarely mutated in AML, the functional activity of Pu.1 is perturbed by several key oncogenic drivers of AML, including AML1-ETO, PML-RARA and MLL-fusion proteins[39–42]. Consistent with recent reports, we found that Lsd1 inhibition results in widespread increased binding of Pu.1[30], but importantly and in agreement with the in silico prediction, we observed that co-occupancy of Pu.1 and Irf8 is far more restricted (Supplementary Fig. 8B) and occurs at the newly formed enhancers that are associated with genes critical to the resensitisation process (Fig. 4f, g).

To investigate the role of Pu.1 in mediating the enhancer switching that resensitises the resistant cells to BET inhibition, we used two independent shRNA's to deplete Pu.1 in the resistant cells (Supplementary Fig. 8C). These data showed that following knockdown of Pu.1, Lsd1 inhibition failed to differentiate and resensitise the resistant cells (Fig. 5a, b, Supplementary Fig. 8D, E). While the expression of Pu.1 is unaltered by Lsd1 inhibition (Fig. 5e), this treatment displaces Lsd1 from the *Irf8* locus leading to marked upregulation of Irf8 (Fig. 5f and Supplementary Fig. 9A). To explore the importance of Irf8, we knocked down this factor and treated the resistant cells with Lsd1 inhibitor (Supplementary Fig 9B). Here we found that although Irf8 depletion failed to prevent Lsd1 inhibitor induced myeloid differentiation, resensitisation was abrogated (Fig. 5c, d and Supplementary Fig 9C, D). In support of this finding, treatment with ATRA fails to resensitise the cells, despite inducing a more differentiated phenotype (Supplementary Fig. 9E, F).

The resensitisation of the resistant cells following Lsd1 treatment is accompanied by the following two major features: (i) the induction of gene expression changes resulting in myeloid differentiation and (ii) the formation of new enhancers. To experimentally dissect the contribution of each of these to the process of resensitisation, we performed RNA-seq on Irf8 depleted resistant cells treated with LSD1i. These data showed that the LSD1i-mediated gene expression changes still occur in the context of Irf8 depletion (Fig. 5g–i and Supplementary Fig. 9G). Consistent with this finding, the LSD1i induced transcriptional changes appear to be the result of direct de-repression of Lsd1 target genes and do not require the Irf8

dependent new enhancers (Supplementary Fig. 10A–D). As Irf8-depleted cells remain resistant to IBET (Fig. 5d), this experiment suggests that the immunophenotypic and transcriptional changes associated with LSD1i mediated differentiation are not sufficient to drive resensitisation of the resistant cells. In contrast to the gene expression changes, depletion of Irf8 is critical for the formation of new enhancers associated with Lsd1 inhibition (Fig. 5j) and the associated resensitisation (Fig. 5d), indicating that it is the enhancer remodeling associated with Lsd1 inhibition that is primarily responsible for resensitisation.

**Pu.1 requires Irf8 to form enhancers during resensitisation.** Pioneer transcription factors are able to engage nucleosomal DNA to establish new enhancers and Pu.1 has previously been demonstrated to actively initiate the de novo formation of enhancers[43]. To understand the precise order of events that orchestrate the development of the new enhancers following Lsd1 inhibition, we first explored the consequences of Pu.1 depletion on enhancer formation. These data show that reduction of Pu.1 abrogates the increased chromatin accessibility and subsequent H3K4me1/2 and H3K27ac suggesting that Pu.1 is the initiating event of new enhancer formation (Fig. 6a, b and Supplementary Fig 10E–G). Consistent with the recent reports of genome-wide motif scanning by pioneer factors[44,45], we see low-level enrichment of Pu.1 at enhancers that become activated upon Lsd1 inhibition (Fig. 6c). However, this low affinity binding is not sustained in the absence of the cofactor, Irf8, which is required to stabilize Pu.1 at chromatin (Fig. 6d). Interestingly, Irf8 over-expression is insufficient to fully initiate new enhancer formation and resensitisation (Supplementary Fig. 10H-J), suggesting that nucleation of these new enhancers requires both the stabilization of Pu.1 at chromatin and inhibition of the recursive enhancer decommissioning by Lsd1. Following Lsd1 inhibition, Pu.1 is stabilized by Irf8 and able to nucleate new enhancers, which are then bound by transcriptional coactivators, including Brd4 (Fig. 6c). The re-distribution of Brd4 to the new enhancers is required to sustain the expression of a set of broadly expressed genes critical to the survival of the cells. As Brd4 is bound at these new enhancers via its bromodomains (Supplementary Fig. 7B), re-challenge with IBET is able to displace Brd4 from the new enhancers (Supplementary Fig. 7C) and repress the transcription of these genes (Fig. 4a). Together, these data provide a detailed molecular understanding of the resensitisation process (Fig. 6e).

**Discussion**
Relapsed AML is frequently an incurable disease, highlighting the urgent need to understand mechanisms of adaptation that result in disease relapse after a clinical remission. A potentially under-appreciated modality of therapeutic resistance is transcriptional

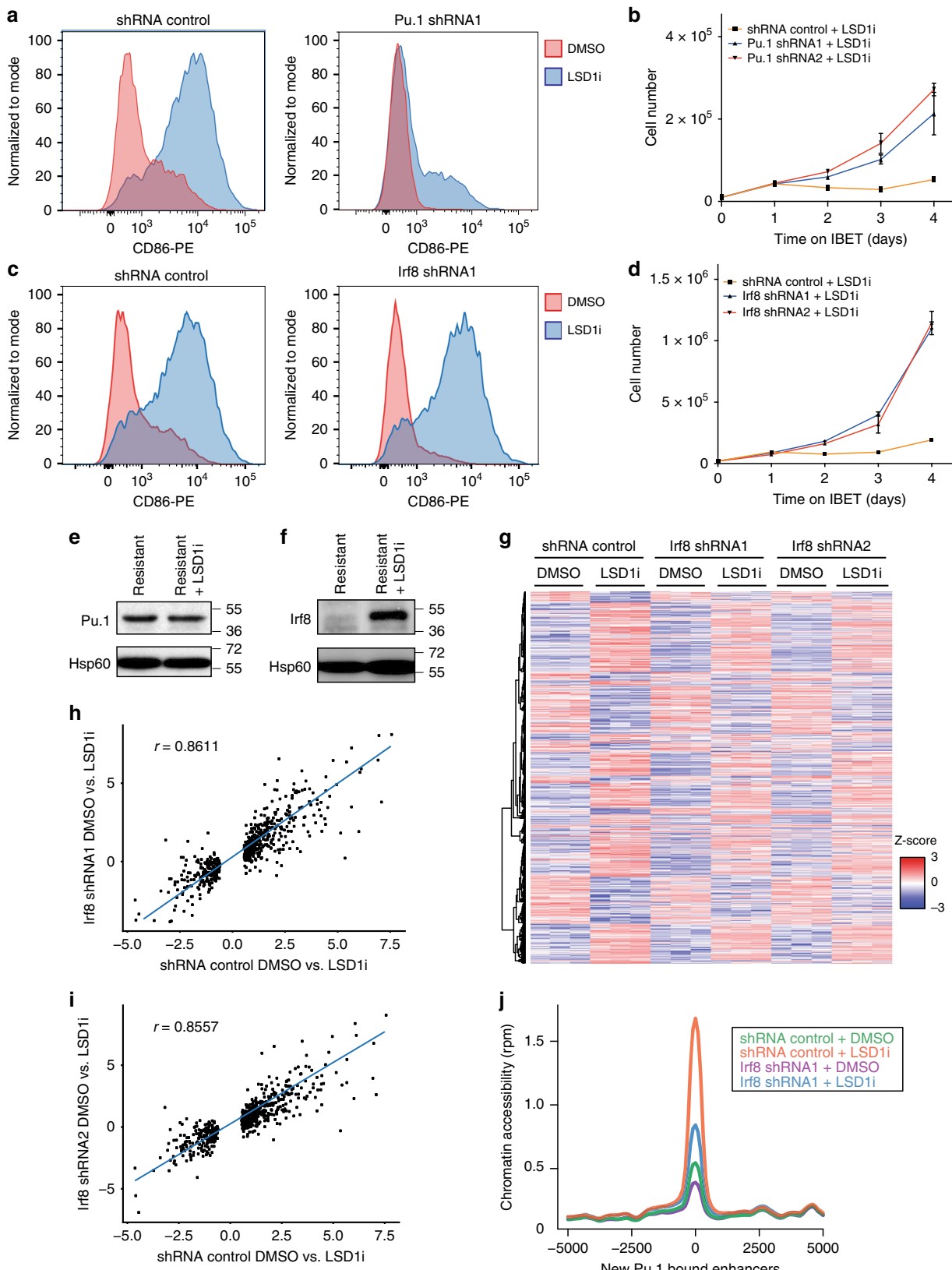

plasticity, whereby malignant cells adaptively rewire themselves to evade a therapeutic challenge[6,46]. While there is now compelling evidence for non-genomic evolution in cancers with a low mutation frequency such as AML, other malignancies can undergo this form of adaptive response to a range of cancer therapies[47,48]. Here we have focused our studies on stable non-genetic resistance to BET inhibitors in AML and have established several important principles that might be applicable to variety of malignant contexts. In contrast to many previous studies of non-genetic resistance, our work has focused on 'stable'

**Fig. 5** Enhancer remodeling mediated by Pu.1 and Irf8 is required for resensitisation, rather than LSD1i-mediated differentiation. **a** Flow cytometry of CD86 expression in the resistant cells expressing shRNA_control and shRNA_Pu.1. The shRNA expressing resistant cells were treated for 6 days with DMSO or GSK-LSD1i (500 nM). **b** Proliferation assay of shRNA_control and shRNA_Pu.1 resistant cells pre-treated for 6 days with GSK-LSD1 (500 nM) followed by treatment with DMSO or IBET (1000 nM). Error bars represent S.E.M of 3 cell culture replicates. Representative of 3 biological replicates. **c** Flow cytometry of CD86 expression in shRNA_control and shRNA_Irf8 cells treated for 6 days with DMSO or GSK-LSD1i (500 nM). **d** Proliferation assay of shRNA_control and shRNA_Irf8 resistant cells pre-treated for 6 days with GSK-LSD1 (500 nM) followed by treatment with DMSO or IBET (1000 nM). Error bars represent S.E.M of 3 cell culture replicates. Representative of 3 biological replicates. **e** Western blot of Pu.1 levels in resistant cells and resistant cells treated for 6 days with GSK-LSD1i (500 nM). **f** Western blot of Irf8 levels in resistant cells and resistant cells treated for 6 days with GSK-LSD1i (500 nM). **g** RNA-seq heatmap displaying the LSD1i associated gene expression changes in shRNA_control and shRNA_Irf8 resistant cells treated for 6 days with DMSO or GSK-LSD1i (500 nM). LSD1i-associated gene expression changes defined as genes that are differentially expressed upon 6 days GSK-LSD1i (500 nM) treatment in shRNA_control resistant cells. **h** Scatter-plot displaying the fold change of the LSD1i associated gene expression changes (defined in **g**) in shRNA_control and Irf8_shRNA1 resistant cells. **i** Scatter-plot displaying the fold change of the LSD1i associated gene expression changes (defined in **g**) in shRNA_control and Irf8_shRNA2 resistant cells. **j** Average profile of chromatin accessibility at the newly formed Pu.1 bound enhancers in shRNA_control and Irf8_shRNA1 resistant cells treated with DMSO or GSK-LSD1i for 6 days

non-genetic resistance. In AML, stable non-genetic resistance appears to be common, as illustrated by clinical experience with relapsed AML patients, our scRNA-seq data from the BET inhibitor clinical trial and our in vitro model. Unlike models of drug persistence, stable epigenetic adaptation cannot be overcome by a drug holiday and instead requires active cellular reprogramming. Our clinical data is derived from a limited number of AML patients with a good partial remission of moderate durability and it is possible that resistance mechanisms in patients with longer remissions or in the context of combination therapies may vary. Nevertheless, it is important to note that the principles established here might have broader application in other cancers and treatment contexts where there is increasing evidence of stable non-genetic mechanisms of resistance.

Here we show that stable non-genetic resistance is not necessarily due to selection of a pre-existing clone but can evolve through adaptive transcriptional plasticity. While this adaptive response is accompanied by widespread transcriptional changes and phenotypic alteration to a more immature cancer stem cell like state, we find overcoming resistance does not necessarily depend on reversing the transcriptional changes associated with the resistant state nor is phenotypic reversion through differentiation sufficient. Instead, our data argues that it is the compensatory mechanism by which cancer cells use alternative enhancers to sustain the expression of a small subset of broadly expressed key survival genes, that is important and targeting this process is the key to overcome non-genetic resistance. Intuitively, the formation of new enhancers without a change in transcriptional output seems like a surprising result; however, enhancer switching is critical in normal development, where stem cells use different enhancers to lineage specific cells to maintain the expression of important broadly expressed genes such as Myc, that are ubiquitously required for cellular homeostasis in virtually all cells[36,37]. Similarly, here an inherently plastic cancer cell uses the available pioneer factors/co-factors to nucleate different enhancers to sustain expression of key survival genes such as Myc. In the model we have studied, the myeloid specific factors Pu.1/Irf8 are critical. In other cell lineages we expect other transcription factors will be important. Our data suggests that rather than aim to inhibit the final transcriptional state/pathway of a resistant cells, which is likely to be a constantly moving target in an inherently plastic cell, it may be more effective to disable the process of enhancer remodelling, which is the cornerstone of non-genetic adaptation and resistance. It raises the prospect that epigenetic therapies may be effective when incorporated into maintenance strategies to curtail the transcriptional adaptation via enhancer remodeling in residual malignant cells often present at clinical remission.

## Methods

**Cell culture**. MLL-AF9 parental, IBET sensitive (drug naïve) and IBET-resistant cell lines were generated previously[6]. MLL-AF9 cell lines were grown in RPMI-1640 supplemented with mouse IL-3 (10 ng/ml), 20% FCS, streptomycin (100ug/ml), penicillin (100 units/ml) and glutamax, under standard culture conditions (5% $CO_2$, 37 °C). MLL-AF9 IBET-resistant cells were cultured with constant IBET treatment (1000 nM, 0.1% DMSO), while MLL-AF9 IBET sensitive cells were cultured with constant DMSO treatment (0.1%). HEK293T cells were cultured in DMEM with 10% FCS with streptomycin (100ug/ml) and penicillin (100 units/ml) in 10% $CO_2$, 37 °C. Human AML cell lines (MV4;11, NOMO-1, THP-1, GF-D8, SKM-1, KG-1 and UT-7) were grown in RPMI-1640 supplemented with 20% FCS, streptomycin (100ug/ml), penicillin (100 units/ml) and glutamax, under standard culture conditions (5% $CO_2$, 37 °C). All cell lines were subjected to regular mycoplasma testing and underwent short tandem repeat (STR) profiling. Human AML cell lines were obtained from ATCC.

**In vitro drug treatment**. DMSO, IBET (IBET-151), LSD1i (GSK-LSD1)[28] and ATRA were dosed by direct addition to the culture media at 0.1%. Drugs were refreshed every two days to ensure maximal activity. IBET was used at a final concentration of 1000 nM. LSD1i was used at a final concentration of 500 nM. ATRA was used at a final concentration of 1000 nM.

**Cell proliferation and dose–response assays**. For dose–response assays, serial dilutions of IBET, were further diluted in media before addition to 96-well plates containing cells, to obtain a 0.1% DMSO final concentration. After 72 h incubation, resazurin was added to each well and plates were further incubated for 3 h. Fluorescence was then read at 560 nm/590 nm on a Cytation 3 Imaging Reader (BioTek). For proliferation assays, cells were seeded at a consistent density prior to treatment in triplicate and treated with DMSO, IBET, LSD1i or both over the indicated time period. Drug was refreshed at least every two days. Cell number was calculated each day using the BD FACSverse (BD Biosciences).

**Flow cytometry analyses**. Flow cytometry analyses were performed on the LSRFortessa X-20 flow cytometer (BD Biosciences). Data were analyzed with FlowJo software (Tree Star). Cell sorting was performed on a FACSAria Fusion flow sorter (BD Biosciences). The following antibodies were used for flow cytometry on cultured cells: AF700 anti-mouse Ly6-G/Ly6C (Gr-1) (108422, Biolegend), PE anti-mouse CD86 (105008, Biolegend), propidium iodide (P4864, Sigma Aldrich), APC-Cy7 anti-mouse CD117 (105826, Biolegend).

Patient bone marrow mononuclear cells (BM-MNCs) were stained with the following antibodies: FITC anti-human CD45 (11-9459-42, eBioscience), Biotin anti-human CD3 (555338, BD Pharmigen), Biotin anti-human CD19 (555411, BD Pharmigen), BV711 anti-human CD38 (563965, BD Pharmigen), APC anti-human CD34 (555824, BD Pharmigen), PE anti-human CD90 (561970, BD Pharmigen), PerCP-Cy5.5 anti-human CD45RA (45-0458-42, In Vitrogen) and V500 streptavidin (BD Horizon, 561419). Propidium iodide was used to select for viable cells and anti-mouse CompBeads (552843, BD biosciences) were used for single-stain controls. Leukaemic blasts were identified using the CD45/SSC gating procedure and CD3/CD19 expression was used to exclude any lymphocytes. LMPP-like LSCs were identified by CD34 + CD38−CD45RA + CD90− expression as previously published[49].

**Chromatin immunoprecipitation sequencing (ChIP-seq)**. Chromatin immunoprecipitation was performed as described previously[50]. Briefly, for each ChIP, 20 million cells were crosslinked for 15 mins with 1% formaldehyde. Crosslinked material was sonicated to ~200–1000 bp using the Covaris Ultrasonicator e220. Sonicated material was incubated overnight with each antibody, then incubated for 3 h with Protein A magnetic beads. Beads were washed with low and high salt wash buffers, LiCl buffer and TE, before being eluted and decrosslinked overnight. DNA

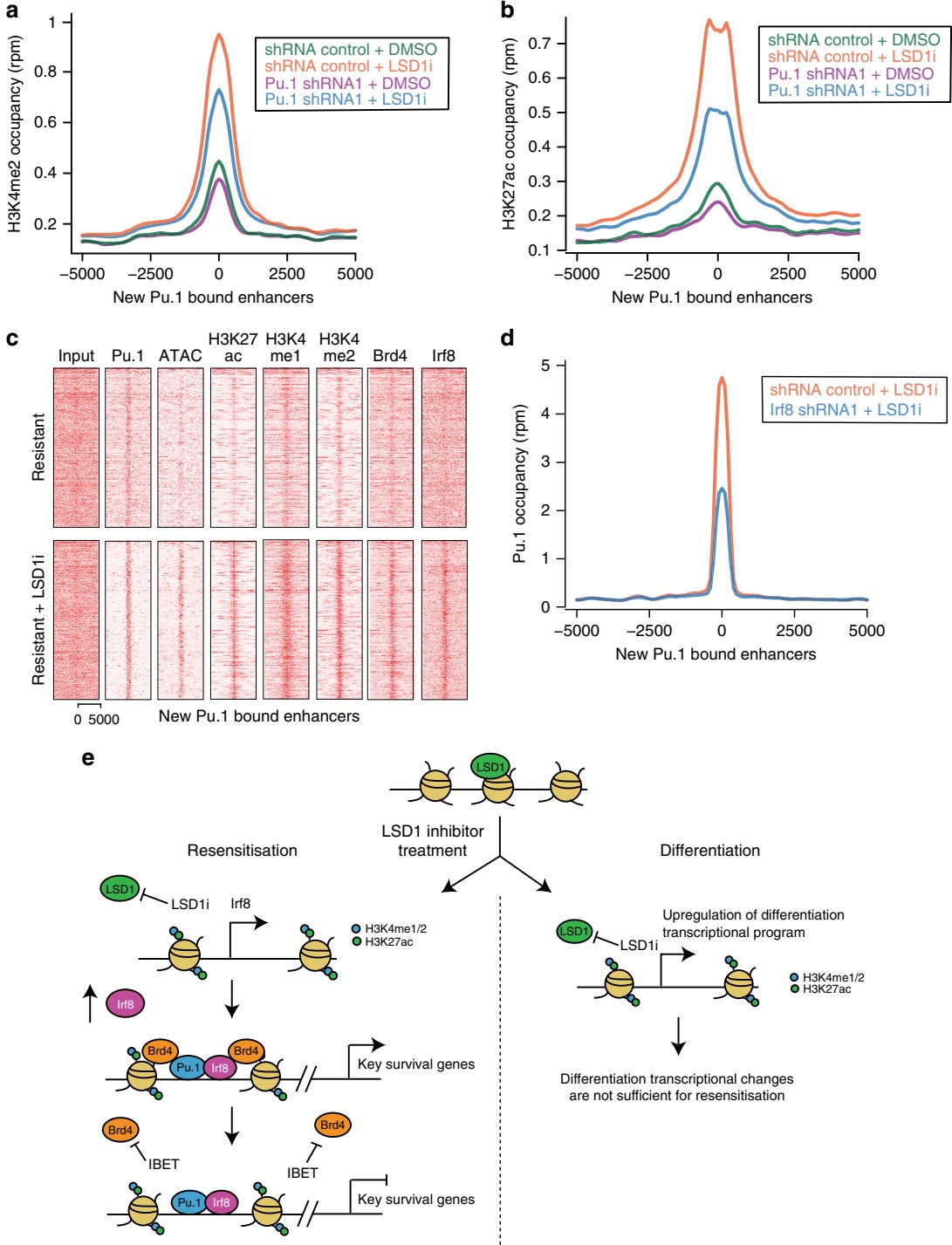

**Fig. 6** Irf8 stabilises low-level Pu.1 occupancy to drive new enhancer formation. **a** Average profile of H3K4me2 occupancy in shRNA_control and Pu.1_shRNA1 resistant cells at sites of increased Pu.1 binding (>4-fold increase) after 6 days treatment with either DMSO or GSK-LSD1i (500 nM). **b** Average profile of H3K27ac occupancy in shRNA_control and Pu.1_shRNA1 resistant cells at sites of increased Pu.1 binding (>4-fold increase) after 6 days treatment with either DMSO or GSK-LSD1i (500 nM). **c** Heatmap of chromatin accessibility and ChIP-seq occupancy for the specified proteins at activated Pu.1 bound enhancers in the resistant cells before and after 6 days treatment with GSK-LSD1i (500 nM). **d** Average profile of Pu.1 occupancy at the newly formed Pu.1 bound enhancers in shRNA_control and Irf8_shRNA1 resistant cells treated with DMSO or GSK-LSD1i for 6 days. **e** Schematic diagram summarizing the divergent molecular mechanisms underpinning differentiation and re-sensitization of the resistant cells after LSD1i treatment

was purified using Qiagen Minelute columns. All ChIP antibodies were used at ~10ug per IP and are listed under the antibodies section. Minor modifications were made for Lsd1 and Brd4 ChIP, including using 60 million cells and crosslinking for 20 mins with 1% formaldehyde. Sequencing libraries were prepared from eluted DNA using Rubicon ThruPLEX DNA-seq kit. Libraries were size selected between 200–500 bps and sequenced on the NextSeq500 using the 75 bp single-end chemistry.

**ChIP-seq analysis**. Reads were aligned to the mouse genome (GRCm38.78) with BWA-mem[51]. Duplicate reads and reads mapping to blacklist regions or the mitochondria were removed. Peak calling was performed with MACS2[52] with default parameters. Genome-browser images of ChIP–seq data was generated by converting the bam files from BWA to TDF files with igvtools and viewing in IGV[53]. ChIP–seq coverage across selected genomic regions was calculated with BEDtools[54].

**New enhancer analysis**. New enhancers were identified by H3K27ac peaks with > 4-fold higher read coverage after LSD1i treatment in IBET Resistant cells within 50 kb of IBET responsive genes (downregulated > 1.5 FC, p value < 0.01 with 6 h IBET in drug naive cells). New Pu.1 enhancers were defined by an increased in Pu.1 occupancy (>4-fold) in resistant cells with LSD1i treatment. New Pu.1 active enhancers were defined by an increased in Pu.1 occupancy (>4-fold) and increase in H3K27ac ( > 4-fold) in resistant cells with LSD1i treatment. Average profiles and heatmaps of ChIP–seq and ATAC-seq reads in the 5 kb around the new enhancers were generated with ngs.plot[55]. Motif analysis on the new enhancers was performed with MEME and CentriMo[56].

**Lentivirus and retrovirus production and transduction**. Lentivirus was prepared by transfecting HEK293T cells with shRNA:pVSVg:psPAX2 plasmids in a 3:2:1 ratio using PEI reagent. Retrovirus was prepared by transfecting HEK293T cells with shRNA:gagPol:pVSVG in a 10:4:1 ratio using PEI reagent. The viral supernatant was collected 48 h following transfection, filtered through a 0.45 µm filter, and added to cells.

**Positive selection CRISPR-Cas9 screen**. CRISPR-Cas9 screens were conducted on a polyclonal population of MLL-AF9 IBET-resistant cells that had been transduced with FUCas9Cherry (Addgene #70182). Highly positive Cas9 cells were sorted to ensure maximal editing efficiency. The epigenetic domain specific sgRNA library (~1200 guides) used in both screens was obtained through collaboration with the Vakoc laboratory[57]. The sgRNA vector was amplified to maintain guide representation. Lentivirus from the sgRNA library was prepared as described above. Sufficient MLL-AF9 IBET-resistant cells were used to maintain 5000 fold representation at all stages of the screening process. The cells were transduced with an appropriate volume of viral supernatant to ensure only a single guide was present in most cells (MOI = 0.3). Transduced cells were sorted and allowed to grow in culture for 7 days. After 7 days, differentiated cells (GR1+) were isolated by flow cytometry. The library control was also isolated at this corresponding time point to provide a reference to determine enrichment due to positive selection. GR1 − enrichment was determined by calculating the depletion of guides in the GR1+ population relative to the library control. Genomic DNA was extracted using DNAse blood and tissue kit (Qiagen), according to the manufacturer's instructions. PCR was conducted to maintain guide representation, using AmpliTaq Gold DNA Polymerase (Thermo Fischer). PCR products were pooled and sequenced on the NextSeq500 using 75 bp paired-end chemistry.

**CRISPR-Cas9 screen analysis**. Sequencing reads containing sgRNA sequences were extracted from fastq file using the linux grep function for the vector sequence "TTGTGGAAAGGACGAAACACCG" immediate 5′ to the sgRNA. The sequences of the sgRNA and barcode were counted using the processAmplicons function in edgeR[58]. RSA[59] was used to identify the genes with sgRNA that were significantly changed in the selected cell populations. Plots of fold enrichment and –log P value of the most differently expressed guide for each gene were generated with ggplot2 in R.

**Syngeneic mouse models of leukemia**. Quaternary syngeneic transplantation studies were performed with intravenous injection of $2 \times 10^6$ MLL-AF9 cells, into C57BL/6 female mice. The cells were obtained from bone marrow or spleen of mice that were injected with cells subjected to serial passaging (4°) in vivo in the presence of DMSO or IBET treatment, as described previously[6]. Briefly, each generation of C57BL/6 female mice were intravenously injected with $2 \times 10^6$ MLL-AF9 cells and treated daily with either DMSO or IBET (20 mg/kg). All mice were inspected daily and sacrificed upon signs of distress and disease. Cells from the bone marrow of each generation were harvested and stored at −80 °C, until they were injected into the subsequent generation. For the quaternary syngenic transplantation experiments included in this manuscript, all mice were 6–8 weeks old at the time of sublethal irradiation at a dose of 3.5 Gy. Treatment with vehicle, IBET, GSK-LSD1 or both began at day 9 via IP daily. IBET was administered at 20 mg/kg/day, and GSK-LSD1 was administered at 0.5 mg/kg/day. All mice were kept in a pathogen-free animal facility, inspected daily and sacrificed upon signs of distress and disease. All studies were conducted in accordance with the GSK Policy on the Care, Welfare and Treatment of Laboratory Animals and were reviewed by the Institutional Animal Care and Use Committee at GSK or were conducted under the approval of the institutional animal ethics review board and were authorized by the Animal Experimentation Ethics Committee (AEEC), Peter MacCallum Cancer Centre. All mice were randomized before the commencement of experiment. Differences in Kaplan–Meier survival curves were analysed using the log-rank statistic.

**shRNA-mediated knockdown**. shRNAs targeting mouse Pu.1, Irf8 and non-targeting control were cloned into the pHR-SIREN lentiviral vector. Kdm1a/Lsd1 shRNAs and the respective non-targeting control were cloned into retroviral MSCV-LMP (Open Biosystems), which has been modified to change GFP into BFP.

**qRT-PCR**. RNA was extracted using the Qiagen RNeasy kit. cDNA was prepared using SuperScript Vilo (Thermo Fischer) according to the manufacturers' instructions. Quantitative real-time PCR was performed on the Applied Biosystems StepOnePlus System using Fast SYBR green reagents (Thermo Fischer). Expression levels were determined using the ΔΔCt method normalised to Gapdh or β2 microglobulin.

**shRNA and primer sequences**. The sequences for shRNAs and primers are included in Supplementary Data 3.

**RNA-seq**. RNA was extracted using the Qiagen RNeasy kit. RNA concentration was quantified with a NanoDrop spectrophotometer (Thermo Scientific). Libraries were prepared using QuantSeq 3′ mRNA-seq Library Prep kit (Lexigen). Libraries were sequenced on the NextSeq500 using the 75 bp single end chemistry.

**RNA-seq analysis**. Reads were aligned to the mouse genome (GRCm38.78) using HiSAT[60] and reads were assigned to genes using htseq-count[61]. Differential expression was calculated using edgeR[58]. Genes with a false discovery rate corrected for multiple testing using the method of Benjamini and Hochberg below 0.01 and a fold-change greater than 1.5 were considered significantly differentially expressed. Principal component analysis was performed on the RNA-Seq data. Gene set testing with ROAST was performed on voom-transformed RNA-seq data. The lymphoid and myeloid differentiation gene signatures were obtained from de Graaf et al 2016[62]. ST_WNT_ BETA_CATENIN_PATHWAY from the GSEA MSigDB C2 curated gene sets was used for the Wnt/B-catenin signature[63]. The LSC gene signature was obtained from Somervaille et al 2009[64].

**ATAC-seq**. ATAC-seq libraries were prepared using the standard ATACseq protocol, as described previously with minor modifications[65]. Briefly, 50000 cells were lysed to isolate nuclei. Isolated nuclei were then tagmented at 37 °C. Tagmented DNA was then amplified into the final ATACseq libraries using KAPA HotStart Ready Mix (2 × ) (KAPA Biosystems). Libraries were size selected from 150 to 700 bp and sequenced on the NextSeq500 using the 75 bp single- or paired-end chemistry.

**ATAC-seq analysis**. Reads were aligned to the mouse genome (GRCm38.78) with BWA-mem[40]. Duplicate reads and reads mapping to blacklist regions or the mitochondria were removed. Peak calling was performed with MACS2[52] using the no lambda and no model settings.

**Western blot**. Cells were lysed using RIPA whole cell lysis buffer and solubilized by brief sonication. Whole cell lysates were then mixed with Laemmli buffer and separated via SDS–PAGE. SDS-PAGE gels were then transferred to PVDF membranes (Millipore) and incubated with primary antibodies (described below) at 1:1000 of the supplied concentration and secondary antibody conjugated with horseradish peroxidase at 1:10000 of the supplied concentration. Membranes were then incubated with ECL (GE Healthcare) and detected on X-ray film (Kodak).

**Antibodies**. The following antibodies were used for western blot and/or ChIP analyses: anti-PU.1/SPI1 (sc325, Santa Cruz Biotechnology, 2258 S, Cell Signaling), anti-KDM1/Lsd1 (ab17721, Abcam), anti-Irf8 (5628 S, Cell signaling), anti-HSP60 (sc13966, Santa Cruz Biotechnology), anti-H3K27ac (ab4729, Abcam), anti-H3K4me1 (ab8895, Abcam), anti-H3K4me2 (ab32356, Abcam), anti-H3K4me3 (ab8580, Abcam), anti-Med1/CRSP1/Trap220 (A300-793A, Bethyl Labs), anti-Brd4 (A301-985A100, Bethyl Labs), goat anti-rabbit IgG (656120, Invitrogen). The concentration that each antibody was used is highlighted in the methods section for ChIP and western blots.

**Patient AML samples ethics and collection**. Patient samples used for analysis were collected as part of a study approved by the Peter MacCallum Cancer Centre human research ethics committee (10/78). Patients described in this study were enrolled in a phase 1 clinical study at the Peter MacCallum Cancer Centre in patients with acute myeloid leukaemia undergoing treatment with the BET inhibitor GSK525762 (14/105) (Clinical trial code: NCT01943851). All patients provided written informed consent. BM-MNCs were separated using Ficoll density gradient. BM-MNCs were then cryopreserved at −80 °C in 90% FBS and 10% DMSO. BM-MNCs samples from baseline, remission and relapse time points from BET001 and baseline and relapse from BET002 were rapidly thawed in a 37 °C water bath. Cells were then transferred to a 50 ml falcon tube and warm IMDM + 20% FBS + DNase1 (0.1 mg/mL) culture medium was added drop wise to the cells. Cells were then washed with PBS + 1% BSA and subsequently processed for flow cytometry.

**Patient targeted amplicon sequencing**. Bulk leukaemic blasts were sorted from patients for targeted-amplicon sequencing. Targeted amplicon deep sequencing (TS) was performed using the 48.48 Access Array™ system (Fluidigm), against a panel of known COSMIC mutations in 54 genes recurrently mutated in MDS and

AML as described previously[66]. Following amplification, products were harvested, tagged with sample-specific barcodes, pooled together and purified using AMPure XP beads. All samples were analyzed in duplicate to control for PCR artifacts. The purified libraries were then sequenced using 150 bp read length on the Illumina MiSeq platform. Sequenced reads were mapped to the human reference genome (version hg19) using BWA-MEM (version 0.7.12) with default parameters. Mutations with at least 20x coverage, a minimum of 5 reads supporting the variant and a mutant allele fraction greater than 1% were retained for further analysis. Variants that were recurrently observed in more than 50% of the samples (representing likely sequencing and/or PCR artifacts) and those with a high global allele frequency (>0.4%) in the 1000 genomes database were flagged and removed from this curated list. Variants in the curated list were then annotated based on their prognostic or functional relevance, as described previously[67].

**Droplet-based scRNA-seq using 10X.** Approximately 6000 leukaemic blasts from each patient sample were sorted directly into separate wells of a 96-well plate, each containing reverse transcription (RT) master mix (50 uL RT Reagent mix, 3.8 μL RT Primer, 2.4 μL Additive A and $H_2O$ to complete 70 μL). After sorting, the volume of each well containing the leukaemic blasts was brought to 90 μL with $H_2O$ and 10 μL RT enzyme was added, mixed well and loaded onto separate channels of the Chromium$^{TM}$ Single-Cell A Chip. Reverse transcription, cDNA amplification and library preparation were performed based on the manufacturers protocol using the Chromium Single Cell 3′ Library & Gel Bead Kit v2 (10X Genomics). Libraries were then pooled at equimolar ratios and sequenced on the NovaSeq 6000 System (2 × 150 PE reads).

**10X analysis.** The raw sequencing base calls were demultiplexed into fastq files and counts (G1k V37) were generated following the Cell Ranger analysis pipeline (10X Genomics). Further analysis was carried out using Cell Ranger R Kit (v2.0.0; 10X Genomics) in R (v3.5).

**Single-cell FACS sorting for Cel-seq2 scRNA-seq.** The MLL-AF9 drug naïve parental cells were dosed with either 400 nM IBET or 0.1% DMSO (vehicle) for 4 days. The resistant cells were withdrawn from IBET four days prior to the experiment. Viable single cells from each treatment group were sorted on the BD FACSAria W, directly into pre-prepared 384-well plates that contained a mixture of oligo (dT) primer, dNTPs and ERCC spike ins, as outlined in the Cel-seq2 protocol[68]. Plate alignment was verified by sorting single BD CompBeads (51-90-9001291, BD Biosciences) coated in a solution of 5 mg/ml horseradish peroxidase (HRP) (77332, Sigma Aldrich), into 8 empty wells containing 1.2 μl HRP substrate (34028, Life Technologies) (well numbers A1, A2, A23,A24, P1,P2, P23, P24). A colour change from clear to blue indicated that the bead had reached the bottom of the well and the plate was correctly aligned[69]. Following the single-cell sort, plates were frozen at −80 °C ready for Cel-seq2 library construction.

**Cel-seq2 library construction.** A modified version of Cel-seq2 was used to generate single cell cDNA libraries. Briefly, the sorted 384-well plates were defrosted on ice and the cells were lysed and the primers annealed at 65 °C. Primers contained a 24 bp polyT stretch, a 6 bp unique molecular identifier (UMI), a cell-specific 8 bp barcode, the 5′ Illumina adaptor and a T7 promoter. Reverse transcription, second strand synthesis and in vitro transcription were performed according to the Cel-seq2[68] with the exception of the cDNA cleanup step prior to IVT, whereby the volume of the Agencourt AMPure XP beads (A63880, Beckman Coulter) was 5% of the total volume of beads used in the original protocol. Following IVT, the aRNA was fragmented, cleaned up and reverse transcribed with a random hexamer primer containing the sequence complimentary to the 3' Illumina adaptor. Fragments containing both adaptors were selected for by PCR using TruSeq Small RNA primers (Illumina) and cleaned up using a 0.7X bead to cDNA ratio. The quality and concentration of the final cDNA library was checked using a High Sensitivity Bioanalyzer (Agilent). Libraries were pooled at equimolar ratios and sequenced on the Illumina Nextseq.

**Cel-seq2 scRNA-seq analysis.** Reads from single-cell RNA-seq libraries were processed in R and Bioconductor using the scPipe package[70] to map reads to the transcriptome, demultiplex reads to individual cells and counts reads mapping to each gene. Low-quality cells and non-expressed genes were identified with the scater package[71] and removed prior to further analysis. Normalisation of the data was performed using the scran package[72]. The tSNE algorithm was used to identify subpopulations of cells and visualise the data[73,74].

**Pseudotime and tSNE analysis of resistance timecourse.** Bcl2fastq (v2.20; Illumina) was used for generating fastq files. scPipe (v1.3.8) pipeline[75] was used in R (v3.5) to process the fastq, align (G1k V37), de-multiplex and generate counts. Monocle (v2.9)[76] was used for normalization and the generation of tSNE (for adaptive timecourse) and single cell trajectory plots using differentially expressed gene set generated from bulk RNA-seq of MLL-AF9 IBET-resistant cells and drug naïve MLL-AF9 cells cultured in DMSO.

**Barcode library construction.** A library of semi-random 60 bp barcode oligonucleotide following the pattern 6(NNSWSNNWSW) was synthesised by Integrated DNA Technologies (Coralville, IA, USA), made double stranded by PCR and subsequently cloned into the MS2-P65-HSF1-mCHERRY backbone at the EspI (BsmBI) site. Pooled ligations were electroporated into Endura ElectroCompetent cells (60242-2, Lucigen) and plated onto agarose plates at an estimated complexity of 300,000 barcodes. All colonies were harvested and purified using NuceloBond Xtra Maxi-prep kit (740414.10, Macherey-Nagel). The barcode pool was subjected to deep sequencing and a reference library of barcodes was generated using quality control measures described previously[77].

**Barcode transduction and methylcellulose experiment.** A total of $5 \times 10^5$ MLL-AF9 drug naïve cells were infected with the barcode virus at a low MOI so that only 5–10% of cells were mCherry positive, ensuring a single integration per cell. mCherry positive cells were sorted 72 h after transduction. A total of $3 \times 10^3$ cells were taken from both the drug naïve and resistant populations expanded for 6 days. A total of $6 \times 10^5$ cells from each aliquot were seeded into 6 ml of methylcellulose (MethoCult GF M3434, Stem Cell Technologies) to ensure 20-fold representation of the original $3 \times 10^3$ barcodes. The methylcellulose was supplemented with either DMSO or 400 nM IBET. Every 7 days, $6 \times 10^5$ cells were re-plated in fresh methylcellulose supplemented with increasing concentrations of IBET (600, 800 and 1000 nM) or maintained in DMSO. Once the cells had been exposed to 1000 nM of IBET, they were maintained in this concentration for a further 4 weeks. At each timepoint, $2 \times 10^6$ cells from both biological replicates were pelleted and frozen for subsequent genomic DNA extraction. This was performed in biological duplicate. The remaining cells from one biological replicate were single cell sorted into 384-well plates and stored at −80 °C ready for scRNA-seq Cel-seq 2 library construction, as described above.

**Barcoding DNA barcode extraction and amplification.** Genomic DNA was extracted from the barcoded cells using the DNeasy Blood and Tissue Kit (69504, Qiagen). Barcodes were amplified in a 2-step PCR using Q5 High-Fidelity DNA polymerase (M0491S, NEB), which first amplified the barcode sequence and then introduced the Illumina P5 and P7 adaptors and 6 bp index sequence to allow for sample multiplexing. The sampling of sufficient template coverage was ensured by parallel reactions in the first round of PCR. A PCR band of 288 bp was checked for all samples on a 2% agarose gel and DNA concentration was quantified on the Qubit. Biological replicates from all time points were pooled at equimolar ratios and sequenced on the Illumina NextSeq 500.

**Barcode analysis.** Sequencing reads from barcoding experiments in fastq format were filtered to keep only those barcodes that (a) showed a correct constant region (5′-CGGATCCTGACCATGTACGATTGACTA-3′) upstream of the barcode, (b) showed the expected barcode pattern (up to 6 repeating units of NNSWSNNWSW), (c) did not contain N residues, and (d) possessed an average phred quality of 30 across the length of the barcode. True barcodes were mapped to the reference library using bowtie v1.2.2[78]. Only exact matches were allowed. Barcode counts were tallied per sample and ranked by proportion. For each sample, barcodes comprising 90% of the total dataset with a representation above 20 at the baseline time point (i.e. the fold coverage per barcode at the beginning of the experiment) were retained. Finally, the number of barcodes comprising 90% of each sample was then sequentially filtered such that those barcodes not present in the subset comprising 90% of the total dataset of the previous replating were filtered out. This stringent filtering, while necessary to limit technical artefacts which may confound interpretation of the data, will remove some low-frequency biological signal and therefore our data likely represent an underestimation of the true number of clones in the resistant population.

**Melanoma patient-derived xenografts (PDX).** In collaboration with TRACE, PDX models were established using tissue from patients undergoing surgery as part of standard-of-care melanoma treatment at the University Hospitals KU Leuven. Written informed consent was obtained from all patients and all procedures involving human samples were approved by the UZ Leuven Medical Ethical Committee (S54185/S57760/S59199) and carried out in accordance with the principles of the Declaration of Helsinki. PDX models MEL006, MEL015 and MEL029 were derived from a female, male and male patient respectively. All procedures involving animals were performed in accordance with the guidelines of the IACUC and KU Leuven and were carried out within the context of approved project applications P147/2012, P038/2015, P098/2015 and P035/2016. Fresh tumor tissue was collected in transport medium (RPMI1640 medium supplemented with penicillin/streptomycin and amphotericin B). Tumor fragments were subsequently rinsed in phosphate-buffered saline supplemented with penicillin/streptomycin and amphotericin B and cut into small pieces of approximately 3 × 3 × 3 mm³. Tumor pieces were implanted subcutaneously in the interscapular fat pad of female SCID-beige mice (Taconic). Sedation and analgesia was performed using ketamine, medetomidine and buprenorphine. After reaching generation 4 (F4), one mouse with a tumor of 1000 mm³ was sacrificed. This tumor was minced followed by dissociation using collagenase I & IV and trypsin. Cells were

resuspended in serum-free DMEM/F12 medium and 250 000 cells were injected in the interscapular fat pad of 8–16-week-old female NMRI nude mice (Taconic).

**Pharmacological treatment of established PDXs.** Mice with tumors reaching 1000 mm³ were started on the BRAF-MEK combination via daily oral gavage. BRAF inhibitor dabrafenib and MEK inhibitor trametinib were dissolved in DMSO at a concentration of 30 and 0.3 mg/mL respectively, aliquoted and stored at −80 ° C. Each day a fresh aliquot was thawed and diluted 1:10 with phosphate-buffered saline. Mice were treated with a capped dose of 600–6 µg dabrafenib – trametinib respectively in 200 µL total volume.

**PDX targeted sequencing.** Amplification and sequencing primers located in Supplementary Data 3. Genomic DNA samples were sequenced (1000x coverage) for a selected panel of genes (Trusight26, Illumina). Additionally, alternative BRAF splicing was determined by sanger sequencing[79] and for amplification events at the BRAF locus, genomic DNA was isolated and quantitative PCR was performed[80]. Known resistance-conferring mutations in MAP2K2 (exon 2 and 3) and AKT3 (exon 4) were detected by PCR and subsequent Sanger-sequencing.

**Reporting Summary.** Further information on research design is available in the Nature Research Reporting Summary linked to this article.

## Data availability

The sequencing data that support the findings of this study has been deposited into the sequence read archive, which is hosted by the National Centre for Biotechnology Information. The GEO accession number is GSE110901. All other relevant data supporting the key findings of this study are available within the article and its Supplementary Information files or from the corresponding author upon reasonable request. The source data underlying Figs. 2e, 3g, 5e, 5f, Supplementary Fig. 4A, 4D and 10H are provided as a source data file. The remaining source data is available from the authors upon request. A reporting summary for this Article is available as a Supplementary Information file.

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

## Acknowledgements

We thank Gisela Arnau, Tim Semple, Jake Carmody and Tim Holloway for advice and assistance with genomic sequencing, Susan Jackson, Kimberly Smitheman and Jacob Rubin for experimental assistance and the members of the Dawson lab for experimental assistance, helpful comments and input into the manuscript. We thank O. Van Goethem for excellent technical assistance; PDX experiments were developed by TRACE with assistance of E. Leucci, L. Rizzotto, D. Thomas, E. Gommé and K. Crevits. The Tru-Sight26 analysis was performed by S. Vander Borght. We thank the following funders for Fellowship, Scholarship and grant support - Leukaemia Foundation Australia senior fellowship and Howard Hughes Medical Institute international research scholarship 55008729 (M.A.D.), CSL Centenary fellowship (S.-J.D.), NHMRC grant 1085015, 1106444 (M.A.D) and 1128984 (M.A.D, S.-J.D.), Snowdome Foundation (P.Y., M.A.D.), Maddie Riewoldt's Vision 064728 (Y.-C.C.), Victorian Cancer Agency (E.Y.N.L., L.M.), CRUK (M.L.B.) and FWO #G.0929.16 N (J.-C.M).

## Author Contributions

C.C.B. and K.A.F.. performed the majority of the experiments, helped develop the overall concept behind the analysis and helped write the manuscript. Y.-C.C., F.R., M.M.Y., D.V., L.L., P.Y., L.G.M., A.R., O.B., H.P.M., T.M.J., M.L.B., N.K., L.T., D.S.T., L.M., J.S., N.P., C.Y.F., O.G. & E.Y.N.L., contributed to data acquisition, analysis and interpretation. B.E.K., & A.D. oversaw the clinical trial. A.T.P., S.M.G., S-J.D., R.S.A., R.G.K., C.R.V., D.L.G., S.H.N., O.G., E.Y.N.L, J-C.M., R.K.P., provided critical reagents, helped supervise the research and contributed to writing the manuscript. M.A.D. developed the overall concept behind the study, supervised the experiments, analysed the data and wrote the manuscript.

## Additional information

**Competing interests:** B.E.K., O.B., H.P.M., R.G.K., A.D., N.K., and R.K.P. are employees of GlaxoSmithKline. M.A.D. has been a member of advisory boards for CTX CRC, Storm Therapeutics, Celgene and Cambridge Epigenetix. The remaining authors declare no competing interests.

