## [Peer Review File · Nature Communications]

REVIEWERS' COMMENTS:

Reviewer #1 (Remarks to the Author):

The authors have further improved their study and added new data, more complete data panels, and better characterizations of their genomics experiments.

The inclusion of the *Irf8* knockdown experiment that shows *Irf8* is dispensable for differentiation/general gene expression phenotype associated with LSDi but critical for conferring IBET sensitization helps support the theory that re-sensitization is driven by enhancer reprogramming. This substantially improves the paper and helps address the question of differentiation vs. enhancer reprogramming raised in earlier reviews. This data is not without its caveats – for one, it's unclear exactly how phenotypically similar *Irf8* kd and wt LSDi treated cells are (most data is still presented with respect to IBET genes only instead of global characterizations) – for example, *Irf8* kd could lead to an 'alternative differentiation' similar to how ATRA differentiation is different, such that neither impact IBET sensitization.

More to the point, the authors still fail to show if IBET treatment is truly mediated through new enhancers. I feel the authors may have missed an opportunity to more directly show the impact of IBET treatment. (in fact, none of the genomics profiling data is performed in LSDi + IBET cells save for an RNA-seq experiment). For example, IBET treatment of LSD1i-sensitized WT and *Irf8* knockdown cells followed by RNAPII or H3K27ac ChIP-seq (or DNase/ATAC/GRO-seq) might have revealed functional targets of IBET at putative enhancers – if enhancers impacted by IBET are enriched near the key survival genes, and a loss of RNAPII or H3K27ac, etc. at their enhancers only occurs in WT and not *Irf8* kd samples – this might provide a convincing display of how LSD1i reprograms IBET's functional targets in an *Irf8* dependent way, and would make for a much more convincing display of how these effects are mediated through specific enhancers.

Minor technical issue: How is Fig. 5G clustered? There is a gene-tree on the left of the panel, but many of the patterns of similar gene expression are NOT grouped together (perhaps an error/mistake?)

Reviewer #2 (Remarks to the Author):

The paper is nicely revised, and the authors make persuasive cases why some of the additional studies are either not feasible or outside the scope of the current report. I have two substantive comments which are more conceptual at this point

1. The responses in the 2 patients who had benefit from BRD4 inhibition are 1) of short duration and 2) not complete, as blasts only transiently went down to 10-20%. I would suggest that the authors reframe both their description of these cases as having short-term partial responses and would suggest that the rapid regrowth of the leukemia fits with the authors' elegant work that these cells are poised for a non-genetic mechanism of resistance. If patients achieve CR to BRD4 monotherapy or of some combination therapy, the mechanism of acquired resistance may be different. this should be delineated.

2. The inclusion of the melanoma data remains of unclear value. The drug, driver mutation and context are different, and the mechanism of resistance is not the same or at least is not investigated to see if similar mechanisms are at play. I would suggest leaving it out. By contrast, showing in AML patients who relapse after classical chemotherapy have enhancer remodeling (even if not LSD1/IRF8-dependent) would markedly increase the importance of this work without that much more detailed insight (e.g. assess K27Ac before and at relapse). it is not essential but would increase the broad relevance of this nice work in the myeloid leukemia context.

Reviewer #3 (Remarks to the Author):

Most issues raised by the reviewers, in particular those related to the conclusions drawn in the current manuscript, have been addressed by the authors. Importantly, they have adequately addressed the conundrum between the induction of differentiation and resensitization with additional analyses supporting the conclusion of enhancer switching as mechanism for epigenetic resistance. In addition, they have toned down on the extrapolation towards other cancers and treatment types and have put this work in a more accurate perspective of what has been described before. The current work convincingly shows adaptive resistance to iBET treatment and the role of enhancer switching in AML.

Minor comments:

Line 181 should be instead of drug "exposure" drug "withdrawal"?

Line 204, not confined to AML or epigenetic therapies

Reviewer #4 (Remarks to the Author):

In the revised version of the manuscript by Bell and colleagues, considerable attention has been put forward to address concerns raised during the previous two rounds of peer review. We appreciate the authors efforts to tone down the language and interpretation in accordance with the data provided in the manuscript. Our concerns have been sufficiently addressed, specifically the additional details included in the figure panels and the removal of the 4C-seq data from the Supplement. In its revised format, this manuscript should be deemed suitable for publication in Nature Communications.

Reviewer #1 (Remarks to the Author):

The authors have further improved their study and added new data, more complete data panels, and better characterizations of their genomics experiments.

We are glad the referee is positive about our latest round of revisions and thank them for their valuable input throughout the review process.

The inclusion of the Irf8 knockdown experiment that shows Irf8 is dispensable for differentiation/general gene expression phenotype associated with LSDi but critical for conferring IBET sensitization helps support the theory that re-sensitization is driven by enhancer reprogramming. This substantially improves the paper and helps address the question of differentiation vs. enhancer reprogramming raised in earlier reviews. This data is not without its caveats – for one, it's unclear exactly how phenotypically similar Irf8 kd and wt LSDi treated cells are (most data is still presented with respect to IBET genes only instead of global characterizations) – for example, Irf8 kd could lead to an 'alternative differentiation' similar to how ATRA differentiation is different, such that neither impact IBET sensitization.

We are pleased that the reviewer agrees that the Irf8 KD data helps lend further support to the several other lines of evidence we have provided to show that enhancer reprogramming and not differentiation underpins the resensitisation process.

We are however a little confused as to why the reviewer believes that an 'alternative differentiation process' driven by Irf8 that may still play a role in the resensitisation. The reviewer may have misunderstood the data. The data presented in the RNA-seq analysis referred to by the reviewer (Fig. 5G) does not just show the IBET responsive genes. It shows **all** of the genes that are affected by LSD1i treatment and demonstrates that these changes still occur in the context of Irf8 KD (Fig. 5G-I). This is why we believe that the differentiation phenotype and the associated gene expression changes induced by LSD1i treatment are not sufficient to account for the resensitisation. In contrast, although Irf8 KD does not alter the gene expression changes (Fig. 5G-I), it does abolish the formation of new enhancers (Fig. 5J) and consequently the resensitisation.

More to the point, the authors still fail to show if IBET treatment is truly mediated through new enhancers. I feel the authors may have missed an opportunity to more directly show the impact of IBET treatment. (in fact, none of the genomics profiling data is performed in LSDi + IBET cells save for an RNA-seq experiment).

We are again a little uncertain here what the reviewer is requesting. If the issue is that the reviewer feels we have only characterised the response of IBET following re-sensitisation with LSD1i treatment with RNA-seq then

unfortunately the reviewer has missed quite a substantial portion of the manuscript.

In the context of LSD1i + IBET we have performed a large amount of ChIP-Seq data (Fig.4 and Supplementary Fig. 7). We have also performed Click-seq data (Supplementary Fig. 7B). Together these data along with the RNA-Seq data show that the reason for the re-sensitisation is that (i) following LSD1i treatment new enhancers form (ii) BRD4 re-distributes from old enhancers and loads at these new enhancers via its bromodomains and (iii) the re-distributed BRD4 bound at the new enhancers regulates gene expression and is sensitive to IBET as treatment with IBET displaces BRD4 from these new enhancers and decreases gene expression leading to death of the malignant cells.

For example, IBET treatment of LSD1i-sensitized WT and Irf8 knockdown cells followed by RNAPII or H3K27ac ChIP-seq (or DNase/ATAC/GRO-seq) might have revealed functional targets of IBET at putative enhancers – if enhancers impacted by IBET are enriched near the key survival genes, and a loss of RNAPII or H3K27ac, etc. at their enhancers only occurs in WT and not Irf8 kd samples – this might provide a convincing display of how LSD1i reprograms IBET's functional targets in an Irf8 dependent way, and would make for a much more convincing display of how these effects are mediated through specific enhancers.

The answer to the reviewer's suggestions is already contained within the manuscript. We have shown throughout the manuscript that the new enhancers do not form in the absence of Irf8. Therefore, the experiments suggested could not be done, as we would not be able to assess whether IBET treatment affects RNAPII or H3K27ac at these new enhancers if they are not present.

Minor technical issue: How is Fig. 5G clustered? There is a gene-tree on the left of the panel, but many of the patterns of similar gene expression are NOT grouped together (perhaps an error/mistake?)

This is clustered by standard hierarchical clustering.

Reviewer #2 (Remarks to the Author):

The paper is nicely revised, and the authors make persuasive cases why some of the additional studies are either not feasible or outside the scope of the current report.

We thank the reviewer for their positive assessment of the revised manuscript.

I have two substantive comments which are more conceptual at this point

1. The responses in the 2 patients who had benefit from BRD4 inhibition are 1) of short duration and 2) not complete, as blasts only transiently went down to 10-20%. I would suggest that the authors reframe both their description of these cases as having short-term partial responses and would suggest that the rapid regrowth of the leukemia fits with the authors' elegant work that these cells are poised for a non-genetic mechanism of resistance. If patients achieve CR to BRD4 monotherapy or of some combination therapy, the mechanism of acquired resistance may be different. this should be delineated.

We agree with the referee and have described the responses in the patients in more detail in accordance with the reviewer's suggestions. Also as requested we have mentioned in the discussion that patients who achieve a longer remission to BET inhibitor therapy or those undergoing combination therapy with BET inhibitors may have different modes of resistance.

2. The inclusion of the melanoma data remains of unclear value. The drug, driver mutation and context are different, and the mechanism of resistance is not the same or at least is not investigated to see if similar mechanisms are at play. I would suggest leaving it out.

Whilst we agree that the focus of the manuscript should be on AML and have adjusted the title to reflect this, we also believe that the melanoma data is an important aspect of the manuscript. These data demonstrate that non-genetic resistance is not just restricted to AML nor is this form of resistance confined to epigenetic therapies. This fact will be of importance and value to the readers and the field. We would therefore like to keep it in the supplementary figures.

By contrast, showing in AML patients who relapse after classical chemotherapy have enhancer remodeling (even if not LSD1/IRF8-dependent) would markedly increase the importance of this work without that much more detailed insight (e.g. assess K27Ac before and at relapse). it is not essential but would increase the broad relevance of this nice work in the myeloid leukemia context.

The experiments suggested by the reviewer are good ones but are not feasible in a short time frame and would require at least another 12 months to complete.

What is being requested is that we analyse several AML patients with RNA-seq, genome sequencing and ChIP-Seq at the time of presentation (prior to therapy), at remission and again at relapse. Serial samples such as these where sufficient viable cells from bone marrow biopsies are available for RNA, DNA and ChIP studies is extraordinarily rare. The scope of these experiments along with the analyses needed would constitute another substantial manuscript in its own right. Therefore, we are glad that the reviewer agrees this experiment is not essential for our manuscript.

Reviewer #3 (Remarks to the Author):

Most issues raised by the reviewers, in particular those related to the conclusions drawn in the current manuscript, have been addressed by the authors. Importantly, they have adequately addressed the conundrum between the induction of differentiation and resensitization with additional analyses supporting the conclusion of enhancer switching as mechanism for epigenetic resistance. In addition, they have toned down on the extrapolation towards other cancers and treatment types and have put this work in a more accurate perspective of what has been described before. The current work convincingly shows adaptive resistance to iBET treatment and the role of enhancer switching in AML.

We thank the reviewer for their feedback during the review process.

Minor comments:

Line 181 should be instead of drug “exposure” drug “withdrawal”?

This has been corrected.

Line 204, not confined to AML or epigenetic therapies

This has been corrected.

Reviewer #4 (Remarks to the Author):

In the revised version of the manuscript by Bell and colleagues, considerable attention has been put forward to address concerns raised during the previous two rounds of peer review. We appreciate the authors efforts to tone down the language and interpretation in accordance with the data provided in the manuscript. Our concerns have been sufficiently addressed, specifically the additional details included in the figure panels and the removal of the 4C-seq data from the Supplement. In its revised format, this manuscript should be deemed suitable for publication in Nature Communications.

We thank the reviewer for their helpful feedback during the review process.